# Inhibition of D-Ala:D-Ala ligase through a phosphorylated form of the antibiotic D-cycloserine

Sarah Batson[1], Cesira de Chiara [2], Vita Majce[1,3], Adrian J. Lloyd[1], Stanislav Gobec [3], Dean Rea[1], Vilmos Fülöp[1], Christopher W. Thoroughgood[1], Katie J. Simmons[4], Christopher G. Dowson[1], Colin W.G. Fishwick[4], Luiz Pedro S. de Carvalho [2] & David I. Roper [1]

D-cycloserine is an antibiotic which targets sequential bacterial cell wall peptidoglycan bio-synthesis enzymes: alanine racemase and D-alanine:D-alanine ligase. By a combination of structural, chemical and mechanistic studies here we show that the inhibition of D-alanine:D-alanine ligase by the antibiotic D-cycloserine proceeds via a distinct phosphorylated form of the drug. This mechanistic insight reveals a bimodal mechanism of action for a single anti-biotic on different enzyme targets and has significance for the design of future inhibitor molecules based on this chemical structure.

[1] School of Life Sciences, University of Warwick, Coventry CV4 7AL, UK. [2] Mycobacterial Metabolism and Antibiotic Research Laboratory, The Francis Crick Institute, NW1 1AT London, UK. [3] Department of Pharmaceutical Chemistry, Faculty of Pharmacy, University of Ljubljana, Aškerčeva 7, 1000 Ljubljana, Slovenia. [4] School of Chemistry, University of Leeds, Leeds LS2 9JT, UK. Sarah Batson and Cesira de Chiara contributed equally to this work. Correspondence and requests for materials should be addressed to L.P.S. de C. (email: luiz.carvalho@crick.ac.uk) or to D.I.R. (email: david.roper@warwick.ac.uk)

The increased incidence and dissemination of microbial resistance to antimicrobials, the lack of incentive for development of these drugs in the pharmaceutical sector and the relative failure of high throughput approaches in the discovery of new antimicrobial targets, all point to a growing crisis in treating infectious diseases[1,2]. It is clear that many of the most highly successful antibacterial agents target multiple activities in bacterial metabolism, resulting in cellular responses leading to cessation of growth or cell lysis[3,4]. This makes the re-evaluation and further exploration of established antimicrobial targets and natural sources of antibiotics that target them, a potentially powerful approach to future antibacterial development.

D-cycloserine (DCS) has been long known to have antibacterial properties and is unusual in respect of multi-targeting, since it is known to inhibit two sequential enzymes in the bacterial cell wall peptidoglycan biosynthetic pathway, leading to the formation of the dipeptide D-alanyl-D-alanine (D-Ala-D-Ala)[5]. DCS targets alanine racemase (Alr), leading to the formation of an aromatized DCS-PLP adduct, which irreversibly blocks Alr activity[6]. In addition, DCS also targets the next enzyme in the pathway, D-Ala-D-Ala ligase (Ddl). DCS inhibition of Ddl was thought to be by simple, competitive and reversible binding to one of the D-Ala binding sites[5] on this enzyme. The bacterial targets of DCS inhibition in mycobacteria have been recently shown by metabolomics to be both Ddl and Alr but predominantly via the former enzyme[7]. DCS is a natural product of *Streptomyces garyphalus* and *S. lavendulae* and is a structural analog of D-Ala. Its clinical use at present is limited to the treatment of tuberculosis (TB), where it is used as a second-line drug for multidrug resistant strains of *Mycobacterium tuberculosis*. Its utility is limited as DCS is also a co-agonist of the *N*-methyl-D-aspartic acid (NMDA) receptor in the brain. DCS binds to the glycine modulatory site of the NMDA receptor and causes adverse side effects, including seizures and peripheral neuropathy[8]. These rare but serious side effects limit its use as an antibiotic to all but the most recalcitrant *M. tuberculosis* infections and preclude its more general use as an antimicrobial agent. However, oral bioavailability, its general efficacy, high gastric tolerance, low rates of resistance and lack of cross reactivity to other anti-TB drugs mean it is still of considerable interest and potential.

Ddl enzymes present an attractive target for further chemotherapeutic investigation because of their essential and universal role in bacterial cell-wall peptidoglycan biosynthesis, which has been a validated target for antibiotics since the discovery of penicillin. Ddls, which belong to the ATP grasp superfamily, use ATP to activate a single D-alanine substrate. Phosphorylation of D-alanine by the bound ATP (Supplementary Fig. 1) produces a transient phosphoryl carboxylate intermediate susceptible to nucleophilic attack by a second D-amino acid. The resulting dipeptide is then incorporated onto the tripeptide chain of the peptidoglycan by the next enzyme in the cytoplasmic phase of the peptidoglycan biosynthetic pathway[9]. Additional interest in Ddl enzymes arises with altered second-substrate specificity of the vancomycin associated Ddl ligases (e.g., VanA, VanB, VanC), which are the central components in the resistance mechanism[9] and have been recently shown to have been present in the ecosystem for millennia[10]. Although the interaction of DCS with Alr has been thoroughly studied, no comparable structural data has yet been obtained to show how DCS interacts with Ddls. We set out therefore to investigate this by elucidation of Ddl structures in complex with DCS and natural ligands to provide further mechanistic insight into the mode of inhibition. Our results reveal the presence of phosphorylated DCS and ADP in both active sites within a dimer. These results are confirmed "off-crystal" using positional isotope exchange and ATPase inhibition assays. Importantly, calculations indicate that DCSP would not bind to the human NMDS receptor, which is the main cause of its toxicity. DCSP-inhibited EcDdlB represents a significant advance on the mechanism-of-action of this clinically used antibiotic and paves for the development of further improved analogs to target Ddl enzymes.

## Results

**Structure of inhibited EcDdlB reveals phosphorylated DCS**. *E. coli* DdlB (EcDdlB) was co-crystallized, as described in methods, with ATP and D-Ala-D-Ala (Fig. 1a), ADP and D-Ala-D-Ala (Fig. 1b), and ATP and DCS (Fig. 1c) and structures determined at sub 2 Å resolution as summarized in Table 1. The overall structure is described in Supplementary Fig 2. Weighted difference maps at 1.65 Å resolution revealed the surprising discovery that phosphorylated DCS was bound in the high-affinity D-alanine binding site (D-Ala1), with the γ-5′-phosphoryl moiety of ATP having been transferred to the 3-oxygen of DCS (Fig. 1c). Continuous strong electron density extended from the DCS oxygen into the phosphoryl group, which was then linked by two magnesium ions to the product ADP. This finding revealed the existence of a new chemical entity, DCSP, within this inhibited EcDdlB structure. The DCSP moiety mimics the structure of D-alanyl phosphate, which is an obligatory intermediate formed by phosphoryl transfer during the first stage of catalysis, prior to condensation with the C-terminal D-Ala of the D-Ala-D-Ala product of DdlB. DCSP exploits most of the interactions that generate the high affinity site for the first D-Ala substrate (D-Ala1). The amino group of DCSP forms a strong hydrogen bond with the carboxylate group of Glu-15, mimicking the critical interaction made by the α-amino group of D-Ala in the first subsite. The ring oxygen (position 1) of DCSP is hydrogen bonded to the backbone NH of Gly-276, in the oxyanion pocket of DdlB, while the adjacent non-protonated ring nitrogen, which occupies the position of the peptide oxygen of the dipeptide D-Ala-D-Ala product mimics the bifurcated interactions with Gly-276 NH and Arg-255 NH1 made by this atom in the product complex. The DCSP phosphate group is hydrogen bonded to Arg-255 (NH1 and NH2), Lys-215 and the amide nitrogen of Ser-150, replicating the interactions observed with the γ-phosphate of ATP and it forms additional links with the adjacent ADP via two coordinated magnesium ions that bridge between the DCSP and ADP molecules.

**Positional isotope exchange shows DCSP formation in solution**. To demonstrate the formation of DCSP by EcDdlB in solution a positional isotope exchange (PIX)[11,12] experiment was setup using [γ-$^{18}$O$_4$]-ATP. Changes in the isotopic composition of the initial [γ-$^{18}$O$_4$]-ATP species over time were monitored by $^{31}$P NMR spectroscopy (Fig. 2). As described in the PIX reaction scheme (Fig. 2a), the reversible transfer of the γ-P group from [γ-$^{18}$O$_4$]-ATP to DCS (**1** and **3**) induces scrambling in the position of the β-γ bridging $^{18}$O and affects the isotopic composition at γ-P and β-P. This is caused by the free rotation of the β-P group of ADP with subsequent positional exchange of oxygens before the reverse reaction takes place (**2**). As a consequence, decrease of the [γ-$^{18}$O$_4$]-P species and concomitant increase of [γ-$^{18}$O$_3$ $^{16}$O]-P species are observed in the NMR spectrum (Fig. 2b and Supplementary Fig. 3a) as well as transfer of the β-γ bridge $^{18}$O to a non-bridging position, resulting in an upfield change of $^{31}$P chemical shift at the β-P position (Fig. 2c). No PIX reaction was observed in control experiments performed in absence of DCS at the same concentration of EcDdlB, confirming that DCS is the specific acceptor of the ATP γ-P group during the phosphate transfer reaction (Supplementary Fig. 4). A rate of 0.36 h$^{-1}$ was obtained for the PIX reaction by fitting the decrease of the [γ-$^{18}$O$_4$]-P

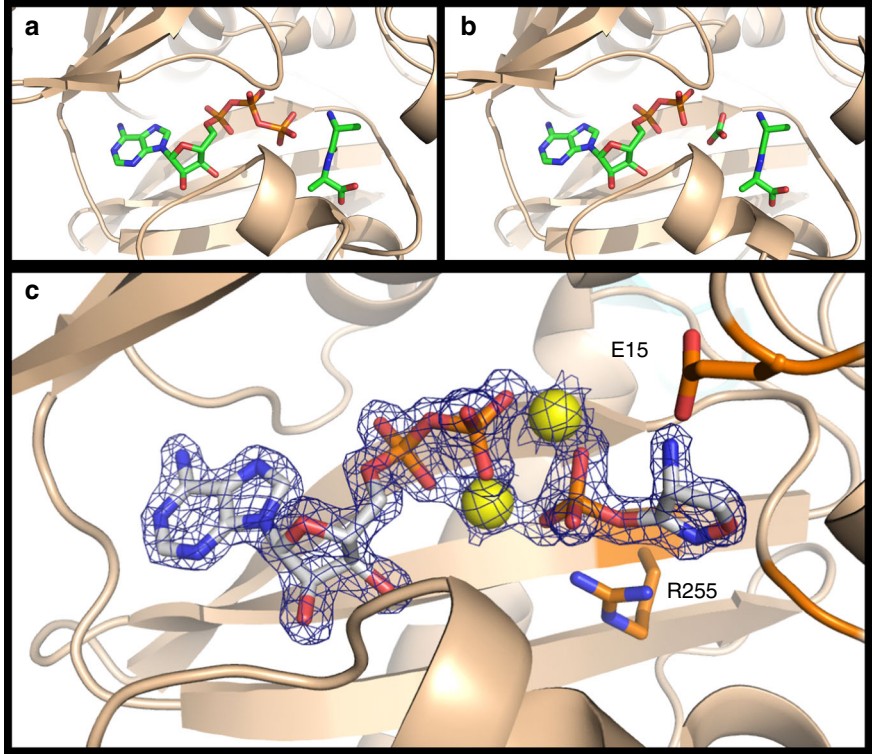

**Fig. 1** Active site region of EcDdlB in different complexes. **a** Active site region of EcDdlB in complex with ATP and D-Ala-D-Ala, and (**b**) ADP, carbonate ion and D-Ala-D-Ala. **c** $2F_o$-$F_c$ difference map of EcDdlB in complex with ADP, 2Mg$^{2+}$ and DCSP. Electron density at 2 $\sigma$ is shown over the ADP, Mg$^{2+}$ and DCSP atoms for clarity. EcDdlB residues Glu15 (above) and Arg255 (below) are within hydrogen bonding distance to DCSP

species fraction (Fig. 2e) to Eq. (1):

$$F = F_0 + Ae^{-kt} \qquad (1)$$

where, $F$ is the fraction of $\gamma$-$^{18}$O$_4$-P scrambled, $F_0$ is the final exchange value of $\gamma$-$^{18}$O$_4$-P, $A$ is the amplitude of change in $\gamma$-$^{18}$O$_4$-P observed during the experiment, $k$ is the observed rate, and $t$ is time. The PIX rate reaches a value of zero at around 8 h. Control experiments (Supplementary Fig. 3b) indicate that this decrease in PIX rate is due to isotopic equilibrium, which as DCS is an inhibitor, or very slow substrate in this particular case, is quite slow. Importantly, as identical PIX kinetics are observed in control experiments, where fresh labeled ATP was added after the equilibrium was achieved (Supplementary Fig. 3b), it is clear that the slowdown in PIX rate is not due to enzyme or substrate inactivation.

**DCS inhibits EcDdlB phosphatase activity.** Under the conditions of the PIX experiment, hydrolysis of ATP was detected as a side reaction over an incubation period longer than 8 h. The phosphatase activity of EcDdlB has been quantified by NMR under the conditions employed for the PIX experiment (Supplementary Fig. 5). In addition, kinetic measurements were performed using a coupled enzyme system (Fig. 2f). Fitting of the initial velocity data from the latter experiment to Eq. (2) provided a $K_m$ for ATP hydrolysis of 167 ± 13 μM and a $V_{max}$ of 0.29 ± 0.01 min$^{-1}$.

$$\nu = \frac{V_{max} \times A}{K_m + A} \qquad (2)$$

where $\nu$ is the velocity at substrate concentration $A$, $V_{max}$ is the maximal velocity, and $K_m$ is the Michaelis constant. Initial velocity data obtained in the presence of DCS show inhibition of ATP

hydrolysis, and were fitted to Eq. (3) (IC$_{50}$ 11.5 ± 0.7 μM), confirming that the observed activity is specific to EcDdlB and not caused by the presence of an adventitious phosphatase contaminant (Fig. 2g).

$$\nu = \frac{\nu_0}{\left[1 + \left(\frac{I}{\text{IC}_{50}}\right)^{nH}\right]} \qquad (3)$$

where, $I$ is the concentration of DCS, IC$_{50}$ is the concentration of DCS necessary to give 50% inhibition, and $n^H$ is the Hill number. Together, these experiments demonstrate a direct interaction of DCS with ATP and positional isotope exchange of ATP γ-P, induced by DCS, consistent with the formation of DCSP.

**The NMDA receptor's glycine site cannot accommodate DCSP.** In relation to neurotoxic effects associated with the binding of D-cycloserine to the NMDA receptor, it has been documented that D-serine, D-cycloserine and glycine bind to the NR1 region of the NMDA receptor in overlapping positions[13]. In order to explore the possibility that the phosphorylated D-cycloserine species may also bind within the NMDA receptor, we applied the docking algorithm eHiTS[14] to the D-cycloserine binding region within the NR1 crystal structure, and modeled the resulting best docking "pose" (Supplementary Fig. 6a–d). These studies indicated that there is insufficient space in the NDMA ligand binding site to accommodate the DCSP species (Supplementary Fig. 6a, b). Using the scoring function in the de novo ligand design program SPROUT[15], the predicted binding affinity for the phosphorylated D-cycloserine to NR1 is around two orders of magnitude lower than that predicted for the binding of D-cycloserine itself to NR1. Importantly, the phosphate group cannot be accommodated within the ligand binding site due to predicted steric clashes of

**Table 1 Data collection and refinement statistics**

|  | EcDdlB DCSP-ADP (PDB: 4C5A) | EcDdlB D-ala-D-ala-ATP (PDB: 4C5B) | EcDdlB D-ala-D-ala-ADP (PDB: 4C5C) |
|---|---|---|---|
| *Data Collection* |  |  |  |
| Synchrotron radiation beamline, detector and wavelength (Å) | Diamond, IO4, ADSC Q315 CCD 0.9763 | Diamond, IO4, ADSC Q315 CCD 0.9763 | Diamond, IO4, ADSC Q315 CCD 0.9763 |
| Unit cell (Å) | $a = 53.79$, $b = 97.51$, $c = 109.99$ | $a = 51.23$, $b = 97.8-$, $c = 110.12$ | $a = 53.00$, $b = 97.62$, $c = 109.49$ |
| Space group | $P2_12_12_1$ | $P2_12_12_1$ | $P2_12_12_1$ |
| Resolution (Å) | 49–1.65 (1.71–1.65) | 49–1.4 (1.45–1.4) | 48–1.5 (1.55–1.5) |
| Observations | 468913 | 648917 | 601481 |
| Unique reflections | 70170 | 106127 | 91443 |
| $I/\sigma(I)$ | 23.8 (2.1) | 22.4 (2.1) | 18.9 (2.1) |
| $R_{sym}$[a] | 0.076 (0.655) | 0.069 (0.577) | 0.093 (0.578) |
| Completeness (%) | 99.8 (99.7) | 97.1 (94.6) | 99.9 (99.1) |
| Refinement non-hydrogen atoms | 5100 (including 2 ADP, 4Mg$^{2+}$, 2 DSC-P, 2 glycerol & 347 water molecules) | 5365 (including 2 ATP, 4Mg$^{2+}$, 2 D-Ala-D-Ala, 3Mg$^{2+}$, 1 imidazole, 2 glycerol & 652 water molecules) | 5483 (including 2 ADP, 4Mg$^{2+}$, 2CO$_3^{2-}$, 2 D-Ala-D-Ala, 2Mg$^{2+}$, 1 imidazole, 2 glycerol & 766 water molecules) |
| $R_{cryst}$[b] | 0.205 (0.318) | 0.180 (0.285) | 0.170 (0.263) |
| Reflections used | 67309 (4861) | 101827 (7180) | 87714 (6275) |
| $R_{free}$[c] | 0.237 (0.327) | 0.203 (0.284) | 0.200 (0.283) |
| Reflections used | 2861 (210) | 4300 (273) | 3729 (258) |
| $R_{cryst}$ (all data)[b] | 0.206 | 0.181 | 0.171 |
| Average temperature factor (Å$^2$) | 16.9 | 11.4 | 11.9 |
| Protein | 16.5 | 10.1 | 10.3 |
| Co-factors | 24.2 | 9.9 | 9.1 |
| Solvent | 20.6 | 20.8 | 21.8 |
| Wilson plot | 19.4 | 12.9 | 13.6 |
| *Rmsds from ideal values* |  |  |  |
| Bonds (Å) | 0.014 | 0.015 | 0.015 |
| Angles (deg) | 1.6 | 1.7 | 1.6 |
| DPI coordinate error (Å) | 0.11 | 0.07 | 0.08 |
| *Ramachandran plot* |  |  |  |
| Most favored (%) | 92.0 | 93.7 | 93.5 |
| Additionally allowed (%) | 8.0 | 6.3 | 6.5 |

[a] $R_{sym} = S_jS_h|I_{h,j} - <I_h>|/S_jS_h<I_h>$ where $I_{h,j}$ is the jth observation of reflection h, and $<I_h>$ is the mean intensity of that reflection
[b] $R_{cryst} = S||F_{obs}| - |F_{calc}||/S|F_{obs}|$ where $F_{obs}$ and $F_{calc}$ are the observed and calculated structure factor amplitudes, respectively
[c] $R_{free}$ is equivalent to $R_{cryst}$ for a 4% subset of reflections not used in the refinement[28]
[d] DPI refers to the diffraction component precision index[30]
The SIGMAA weighted $2mF_o$-$\Delta F_c$ electron density[31] is contoured at the 1.0 $\sigma$ level, where $\sigma$ represents the rms electron density for the unit cell
Numbers in parentheses refer to values in the highest resolution shell

DCSP with the wall of the NMDA ligand-binding cavity (Supplementary Fig. 6c, d).

## Discussion

DCSP is clearly stable under conditions employed to crystallize and diffract EcDdlB but previous work indicated that alanyl-phosphates and related compounds are likely to break down rapidly in aqueous solution, precluding their experimental isolation for confirmatory purposes[16]. The highly reactive nature of acyl phosphates, and in particular, the great difficulties in terms of isolating and characterizing them, has been well established. Although very early chemical approaches have shown that insoluble silver salts of acetyl phosphate itself can be isolated[17] all other reports involving these systems indicate that they are reactive and subject to ready decomposition. For example, under neutral conditions (pH = 7.2), it has been shown that acyl phosphates are rapidly hydrolyzed[18] (e.g., rate of hydrolysis of acetyl phosphate at 39 °C and pH 7.2 = $4.4 \times 10^3$ min$^{-1}$). More recent studies[19] report that acyl-phosphorylated amino acids (e.g., that derived from valine), generated in aqueous solution, can be detected using NMR, but are somewhat transient, and undergo steady decomposition (rate = $5.7 \times 10^{-4}$ s$^{-1}$, corresponding to a halving of the concentration of the acyl phosphate intermediate after 30 min). These authors briefly investigated the mechanism

of decomposition of the acyl phosphates and report that addition of methanol to the aqueous solution followed by evaporation of the solvents allowed the identification of methanol cleavage products (methyl phosphate) corresponding to attack of the phosphate group in the intermediate amino acid acyl phosphate by methanol.

PIX is the method of choice for mechanistic investigation of enzymatic reactions involving phosphate transfer from reactant to product or phosphate exchange between them at equilibrium[20]. The method exploits $^{31}$P NMR to monitor shift of the $^{31}$P resonance by substitution of bonded $^{16}$O with $^{18}$O and has been widely applied to clarify mechanisms of enzymatic reactions, including the demonstration of formation of the intermediate D-alanyl phosphate (also not stable in aqueous medium), during peptide bond formation catalyzed by Enterococcal VanA[16] and Salmonella DdlA[21] (Supplementary Fig. 2). It should be noted that in the case of the experiment with EcDdlB, ATP and DCS the PIX rate is of the order of 0.36 h$^{-1}$, which is significantly slower than the rate obtained with VanA, about < 1 min$^{-1}$[16]. We believe this difference can be attributed to the intrinsic reactivity of the substrate (D-Ala) vs. the inhibitor (DCS).

The phosphatase activity of EcDdlB detected as a side-reaction alongside the PIX reaction was proved to be inhibited by DCS. The IC$_{50}$ obtained for DCS-inhibition of ATPase activity (11.5 µM) is well in the range of $K_i$ values obtained for DCS-inhibition

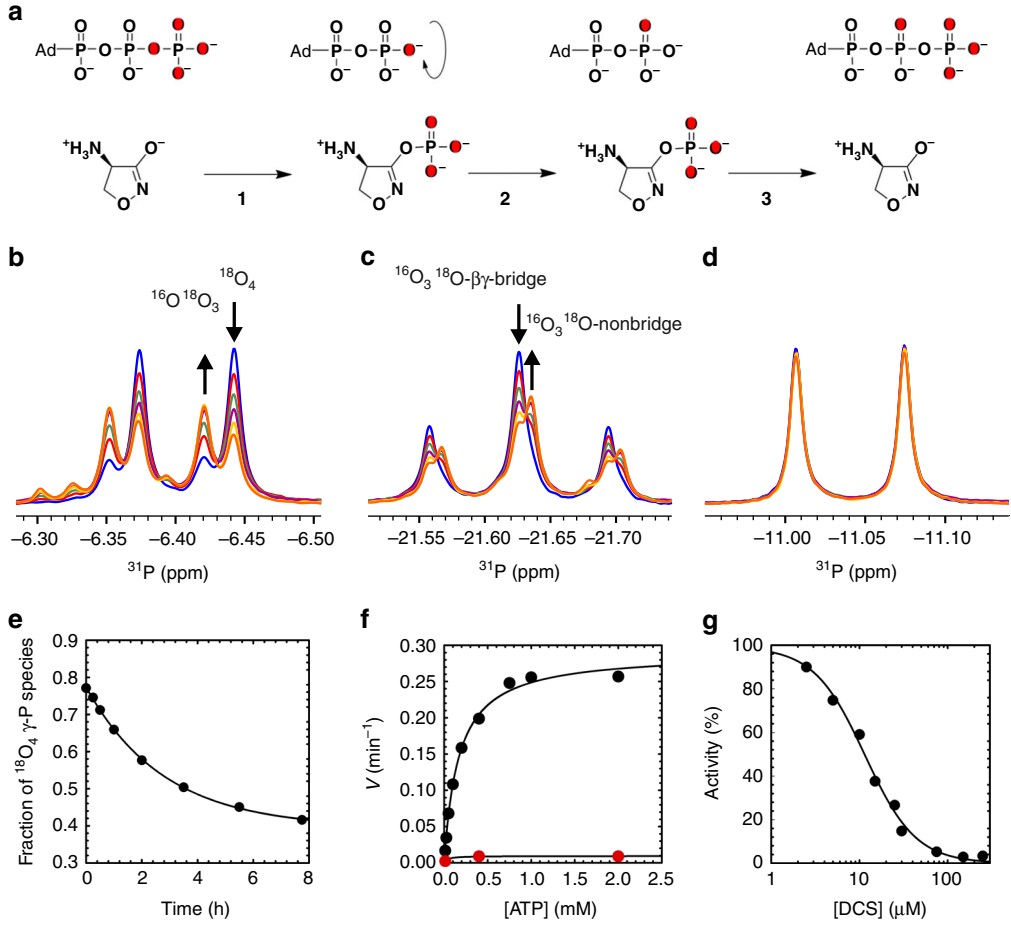

**Fig. 2** Mechanism of DCS phosphorylation by EcDdlB as proved by positional isotope exchange (PIX) and steady-state kinetics of ATP hydrolysis and its inhibition by DCS. **a** Positional isotope exchange (PIX) mechanism for inhibition of EcDdlB by DCS. Position of $^{18}O$ label in the initial $[\gamma-^{18}O_4]$-ATP, intermediate and final species is highlighted in red. **b** $^{31}P$ NMR spectra monitoring changes in the isotopic composition of γ-P (doublet), (**c**) β-P (triplet) and (**d**) α-P (doublet) species of $[\gamma-^{18}O_4]$-ATP as a function of time: (blue) 15 s, (red) 0.5 h, (green) 3.5 h, (purple) 5.5 h, (yellow) 8 h, (orange) 18 h. In accordance with the reaction scheme reported in **a**, a PIX effect is observed both at γ-P and β-P position whereas, as expected, α-P remains unaffected. Based on the relative peak intensity the initial fraction of the $[\gamma-^{18}O_4]$-P species (blue, upfield doublet) and $[\gamma-^{18}O_3 \; ^{16}O]$-P species (blue, downfield doublet) is 77% and 23%, respectively, consistent with a > 94% isotopic enrichment of each of the 4 oxygens. **e** Fractional occurrence of the $[\gamma-^{18}O_4]$-P species during the PIX reaction as a function of time. After 8 h upon addition of the enzyme at 25 °C, the fraction of the $[\gamma-^{18}O_4]$-P species decreased to 42%. The rate constant was obtained from fitting of the data to Eq. (1). **f** Phosphatase activity of EcDdlB (black) as a function of ATP concentration and its inhibition by 1 mM DCS (red) were monitored by a coupled enzyme system assay (See Methods for details). Points are experimental data and lines best fit to Eq. (2). **g** Inhibition of EcDdlB hydrolysis of ATP by DCS. Data were fitted to Eq. (3)

of *E.coli* Ddls biosynthetic reaction (9 μM for DdlA and 27 μM for DdlB)[22]. Interestingly, a value of 1.5 for the Hill number seems to suggest the existence of positive cooperativity for DCS inhibition of ATP hydrolysis reaction.

We conclude that it is highly unlikely that DCSP or analogs developed from this structure, if able to cross the blood-brain barrier, would result in NMDA receptor activation and the side effects associated with DCS treatment. Furthermore, in the high-level vancomycin resistance mechanism found in *Enterococci*, chromosomally encoded Ddl enzymes are superseded in function by transposon-borne D-alanine:D-lactate ligases (e.g., VanA, VanB) that enable the formation of peptidoglycan peptide termini that have low affinity to glycopeptide antibiotics[9,23]. If such DCSP-based inhibitors also bind to the D-alanyl-D-lactate ligases (e.g., VanA, VanB) then this would lead to renewed therapeutic utility of glycopeptide antibiotics in life threatening glycopeptide resistant infections.

In summary, the structural and biochemical analysis of EcDdlB in complex with DCSP represents both the first of a Ddl enzyme in complex with a D-alanyl-phosphate mimic and the only

structure of a Ddl enzyme from any organism in complex with a clinically used antibiotic. Furthermore, our structural studies have suggested a novel mode of action for DCS, involving its phosphorylation to yield DCSP, which is to the best of our knowledge a previously undescribed chemical entity. The results underscore the biochemical flexibility of this remarkably simple antibiotic. This mechanistic diversity of antibiotic action is unusual amongst the natural antibiotics.

## Methods

**Protein purification and crystallization**. Recombinant EcDdlB with an amino-terminal histidine-tag was expressed and purified from *E. coli* BL21(λDE3)Star-pRosetta, crystallized and cryoprotected for data collection as previously reported[24]. Briefly, after cells harvesting, resuspension in buffer (50 mM sodium phosphate, 300 mM NaCl, 10 mM imidazole), lysis by sonication and centrifugation, EcDdlB was purified from the supernatant by Ni-affinity and size-exclusion chromatography.

Crystals were obtained by co-crystallization at 291 K using the hanging-drop method and a 2 μl drop containing one to one ratio of protein and crystallization solution (200 mM MgCl₂, 25% PEG 3350, 100 mM Tris–HCl pH 8.0). Protein concentration was 12 mg ml⁻¹ in 50 mM HEPES pH 7.5, 150 mM KCl, and 0.5 mM EDTA buffer, containing 5 mM ATP or 5 mM ADP and 50 mM D-alanyl-D-

alanine, and 5 mM ATP and 5 mM DCS. Crystals were flash cooled in liquid nitrogen using reservoir solution containing 30% glycerol as a cryoprotectant and used for X-ray diffraction data collection on beamline IO4 at the Diamond Light Source, U.K.

All crystals belonged to the orthorhombic space group $P2_12_12_1$, with two molecules in the crystallographic asymmetric unit[24]. Following structure solution by molecular replacement, refinement of each structure was carried out by alternate cycles of REFMAC[25] using non-crystallographic symmetry (NCS) restraints and manual rebuilding in O[26]. Water molecules were added to the atomic model automatically by Arp/wARP[27] and in the last steps of refinement all the NCS restraints were released. A summary of the data collection and refinement statistics is given in Table 1. For each structure, all 306 amino acid residues of the native protein, along with an extra 8 residues from the N-terminal His$_6$ affinity tag sequence, are visible in molecule 1 of the asymmetric unit and the electron density is of a high quality throughout this molecule (Supplementary Fig. 1). Residues 3–306 could be fitted in molecule 2, but both the N- and C-terminal domains display higher flexibility in this molecule, as evidenced by less well-defined electron density. Nevertheless, the bound ligands are clearly defined in both subunits and refine well.

**Structure description**. Two of the EcDdlB structures solved in this study contained the dipeptide product D-Ala-D-Ala, but differ in the presence of either ADP or ATP. In both these structures, the dipeptide product makes essentially identical interactions with the enzyme. The protonated[28] α-amino group of the N-terminal D-Ala (D-Ala1) makes a salt bridge with the carboxylate group of Glu-15, as in the original structure, to form the high affinity alanine binding site seen in EcDdlB and other related ligases[9,29]. The carbonyl oxygen of the peptide bond makes two hydrogen bonds, with the backbone amide NH of Gly-276 and the side chain of Arg-255 in the oxyanion pocket of the ligase. The C-terminal D-Ala of the product (D-Ala2) is bound with its carboxylate occupying the low affinity alanine binding site centered on side chain hydroxyl of Ser-281 and ε-amino group of Lys-281. The side chain hydroxyl of Tyr-216 also interacts with the product at the backbone NH of D-Ala2, corresponding to its amino end. The adenine ring of the ATP or ADP ligand lies in a pocket mainly composed of hydrophobic and aromatic residues, while the α- and β-phosphate groups are hydrogen bonded to the ε-amino groups of Lys-97, Lys-144, and Lys-215. In the ATP-containing structure, there are further links from the γ-phosphate to Lys-215, Arg-255 and the backbone amide nitrogen of Ser-150 (Fig. 1a). In the ADP-D-ALa-D-Ala structure, we find density for a putative carbonate ion at the equivalent spatial position the γ-phosphate group of ATP would occupy. We initially considered whether this density could correspond to an yttrium ion from the yttrium chloride used in crystallization, but density clearly represents a planar molecular species, not consistent with a metal ion. This was included in the model, and refined, and found to make interactions identical to those made by the γ-phosphate of ATP. The active site interactions observed in the DdlB:ADP:D-Ala-D-Ala structure parallel those observed in the two previous EcDdlB structures with ligands, which essentially mimic the transition state of the EcDdlB reaction (Fig. 1b).

**Kinetic measurements**. All chemicals were of analytical grade and purchased from Fisher Scientific (Loughborough, UK) or Sigma (Poole, UK). [γ-$^{18}$O$_4$]-ATP (>94% isotopic enrichment and 97% chemical purity) was purchased from Cambridge Isotopes Laboratories.

**$^{31}$P NMR experiments**. $^{31}$P NMR spectra were acquired at 25 °C using a Bruker Avance III HD spectrometer equipped with a 5mm quadruple-resonance PFG cryoprobe and operating at a $^{31}$P frequency of 283.4 MHz.

**PIX reaction**. Reaction mixture for PIX contained 100 mM HEPES (pH 7.5), 2 mM [γ-$^{18}$O$_4$]-ATP, 6 mM MgCl$_2$ and 1 mM DCS. The PIX reaction was started by EcDdlB enzyme addition at 24.5 μM and the sample incubated at 25 °C. At time points equal to 15 s, 15 and 30 min, and 1, 2, 3.5, 5.5, 7.75, 18, and 24 h, aliquots of 500 μl were removed and quenched by addition of 50 μl of 0.5 M EDTA (Fig. 2e). Time points for the control experiment (Supplementary Fig. 3b) were taken at 15 s, 30 min, 1.5, 3.5, 6, and 20 h. After collection of the aliquot at 20 h, further 2 mM of fresh [γ-$^{18}$O$_4$]-ATP was added to the reaction mix and time points were taken at 15 s, 30 min, 1.5, 3.5, 6, and 20 h after addition. The final volume of the NMR sample was 600 μl containing 8% D$_2$O added after quenching the reaction. A control experiment for detection of ATP hydrolysis under identical conditions of the PIX experiment was run on unlabeled ATP.

**Phosphatase activity and inhibition assays**. Initial velocities of the EcDdlB phosphatase activity were monitored continuously at 25 °C by UV–Vis spectrophotometry (Shimadzu UV-2550, Milton Keynes, UK) using a 1 cm path-length cell and coupling ATP hydrolysis to NADH oxidation ($\varepsilon_{340} = 6220$ M$^{-1}$ cm$^{-1}$) via a pyruvate kinase and lactate dehydrogenase system. Reaction mixtures contained 50 mM HEPES (pH 7.5), 10 mM MgCl$_2$, 80 mM KCl, 0.10 mM NADH, 1.5 mM PEP, 1 μl ml$^{-1}$ pyruvate kinase/lactate dehydrogenase enzyme solution (PK/LDH; stock solution of 6–10 U ml$^{-1}$ PK and 9–14 U/ml$^{-1}$ LDH) and variable ATP concentration (12.5, 25, 50, 100, 200, 400, 750, 1000, 2000 μM). When included in the

experiments, DCS was present at 1 mM (Fig. 2f). EcDdlB was at a final concentration of 14.4 μM. In the phosphatase inhibition assay (Fig. 2g) DCS concentration was 2.5, 5, 10, 15, 25, 30, 75, 150, 250 μM, whereas ATP concentration was 2 mM.

**Data availability**. The atomic coordinates and the structure factors for *E. coli* EcDdlB in complex with DCSP and D-ala-D-ala peptides have been deposited with the accession codes 4C5A, 4C5B and 4C5C. Other data are available from the corresponding authors upon reasonable request.

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

## Acknowledgements

Crystallographic data were collected at beamline IO4 at Diamond Light Source, U.K., and we acknowledge the support of Dr Ralf Flaig. This work was supported by a University of Warwick PhD studentship to SB and in part by MRC research grants G500643, G0701400, G1100127, Wellcome Trust equipment grants 071998, 068598; the Francis Crick Institute, which receives its core funding from Cancer Research UK (FC001060), the UK Medical Research Council (FC001060), and the Wellcome Trust (FC001060); the Slovenian Research Agency (Grant No. L1-6745 to S.G.); and a UNESCO–L'Oréal International Fellowship "For Women in Science" to V.M. Equipment used in this research was obtained, through the Birmingham-Warwick Science City Translational Medicine: Experimental Medicine Network of Excellence project, with support from Advantage West Midlands (AWM). We thank the MRC Biomedical NMR Centre (The Francis Crick Institute, London) for the use of spectrometers and Dr Geoff Kelly for advice on NMR data collection. We thank Dr Karen Hinxman for help with data deposition.

## Author contributions

S.B., C.d.C., V.M., D.R., C.T., and K.J.S. performed all the experimental procedures. S.G., A.J.L., C.G.D., V.F., and C.W.G.F. supervised biochemical, crystallographic, and modeling experiments respectively, L.P.S.d.C., A.J.L., and D.I.R. wrote the paper and directed the overall research programme.

## Additional information

**Competing interests:** The authors declare that they have no competing financial interests.

