## [Peer Review File · Nature Communications]

Reviewers' comments:

Reviewer #1 (Remarks to the Author):

The manuscript by Batson et al describes, by structural analysis and demonstration of trans-phosphorylation using the technique of positional isotope exchange, that the antibiotic, D-cycloserine, is phosphorylated by MgATP in the active site of the D-ala D-ala ligase (DdlB) of *E. coli*. The authors maintain that this is the first demonstration of this mode of inhibition by an antibiotic. The paper also suggests that the product, D-cycloserine phosphate (DCSP) should not bind to the NMDA receptor, as does D-cycloserine (DCS), the latter of which is neurotoxic. The unequivocal demonstration of DdlB-catalyzed formation of DCSP from MgATP, along with the attending structure DdlB-MgADP-DCPS, is an important finding, as it will influence the design of other "phosphorylatable" analogues of DCS.

However, this reviewer does not believe that this paper is suitable for publication in *Nature Communications* because there is no other detail about whether or not the phosphorylation of DCS has any kinetic significance in the inhibition of DdlB or that its phosphorylation creates a chemical species that is more inhibitory than DCS.

(1) For example, the apparently reversible formation of DdlB-MgADP-DCSP is far slower than D-Ala-D-Ala formation, that is to be expected, but does DCSP formation lead to time-dependent inhibition of DdlB on the same timescale as the rate of PIX, meaning that DCSP leads to a tighter-binding inhibitor than DCS? Or is PIX just an artifact of a long incubation of MgATP, DCS and very large (μM) levels of enzyme?

(2) The inhibition data shown in Figure 1g indicates an IC_{50} value of DCS-mediated inhibition of the ATPase activity of DdlB of $11.5 \mu\text{M}$. How does this value compare with the competitive inhibition constant of DCS vs. D-ala under full biosynthetic reaction conditions? Is the rate of PIX that reaches a value of zero at 8 hrs due to the achievement of PIX equilibrium or has the formation of a tight-binding DdlB-MgADP-DCSP complex shut down the enzyme? If the value is the same at short incubation times with the full complement of substrates, then this suggests that the formation of DCSP provides no inhibitory benefit. It would be helpful to know if the inhibition of DdlB by MgATP + DCS and by MgADP + DCSP are comparable, or does the enzyme-catalyzed formation of DCSP provide an especially potent inhibition as seen for glutamine synthetase + MgATP + methionine sulfoximine.

(3) Is the formation of DdlB-MgADP-DCSP reversible, and thereby, is DdlB expected to create a pool of intra-bacterial DCSP that would perhaps trap DCSP within the bacterium and lessen the availability of DCS to exposure to neural NMDA receptor? This is a simple dialysis experiment to determine if DdlB activity may be restored after the formation of the DdlB-MgADP-DCSP ternary complex.

(4) To point 3, if DCSP is metabolically insignificant, I do not see the value of the modelling studies demonstrating that DCSP should not bind to NMDA. And also, if the course of next-generation inhibitor design pursues the notion that analogues of DCSP are the path to proceed, it should first be shown that DCSP itself is a DdlB inhibitor. This has not been shown. DCSP could be made in micromolar amounts enzymatically.

(5) There is no description of crystallographic conditions for the DdlB-MgADP-DCSP that are salient, at least to this reviewer.

(6) This reviewer believes that eq 1 is meant to be the Michaelis-Menten equation

Reviewer #2 (Remarks to the Author):

In the manuscript entitled "Antibiotic inhibition of D-Ala:D-Ala ligase through a novel phosphoryl intermediate" by Batson et al. (Manuscript NCOMMS-16-28288-T) the authors describe a novel chemical entity, D-cycloserine-phosphate (DCSP) as inhibitor of E. coli D-alanine ligase. The observation of DCSP is a distinct phosphorylated form of D-cycloserine (DCS) generated by the transfer of the γ -5'-phosphoryl moiety of ATP to the 3'-oxygen of DCS. Later molecule is a commercial drug against multi-resistant M. tuberculosis strains.

The authors determined three crystal structures: DdlB co-crystallised with ADP and DdlB co-crystallised with D-Ala, ATP as well as DdlB co-crystallised with D-Ala-D-Ala, and ATP and DCS. The structural investigations are further supported kinetic measurements in particular ³¹P NMR experiments to prove the existence of DCSP.

To the reviewer's point of view the current version of the manuscript is very well written and comprehensive.

- What is the sequence identity between E. coli and M. tuberculosis? Are key residues in the active site conserved or different?
- Do the authors have an idea about the origin of the carbonate in vicinity of ADP and D-Ala-D-Ala?
- Are there differences in the active sites of both monomers in the dimer?

Minor points:

- Figure 1: Why the authors choose a different color for the adenine and ribose function in panel b and c? Maybe the Mg²⁺ ions could also be displayed in panel a and b.
- SI Table 1: It would be beneficial to report CC1/2 for the experimental data
- Frequently spaces are missing between value and unit.

Reviewer #3 (Remarks to the Author):

This report describes a re-evaluation of the mechanism of an old natural product antibiotic, D-cycloserine (DCS), on one of its molecular targets, D-Alanine-D-Alanyl Ligase (Ddl), an enzyme essential for bacterial peptidoglycan biosynthesis. The first reports of DCS inhibition of Ddl dates to 1964 in pioneering work by Neuhaus. DCS also inhibits the preceding enzyme in the biosynthetic pathway, Alanine racemase. Blocking more than one biosynthetic step makes DCS an attractive clinical candidate. Unfortunately, DCS also targets the NMDA receptor, resulting in challenging toxicity issues and diminishing DCS's potential as a useful drug (though it does find use as a back up drug for Mtb infection).

The authors demonstrate, using 3D-structure determination and kinetic PIX experiments, that DCS can be phosphorylated by DdlB, generating a new phosphorylated species. Phospho-DCS is expected (but not explicitly shown) to be unstable, and finding it in the enzyme active site, associated with residues known to stabilize alanine-phosphate, suggests an active site environment that can accommodate phosphor-DCS-like compounds. Using modeling, the authors predict that this phosphorylated species should not bind to the NMDA receptor and therefore new chemical species that mimic this structure should retain enzyme inhibition (and therefore perhaps antibiotic?) activity while diminishing NMDA-associated toxicity.

The finding that DCS is phosphorylated is I believe completely unexpected and unprecedented. This antibiotic has been studied in academic and pharma labs for over 50 years with no hint that this

phospho-DCS was a potential intermediate. The crystal structure is unambiguous and the PIX data are consistent with the proposed mechanism. Overall this is an important addition to the antibiotic literature.

How hard did the authors try to isolate phospho-DCS? I expect while the half life may be fast, it should be detectable.

I am puzzled by some of the kinetic data. The PIX rate was determined to be 18.5 min⁻¹. I'm not aware of the D-Ala PIX rate for DdlB, but the rate for Van enzymes is <1 min⁻¹ (Healy et al (2000) Chem Biol 7:505-14). The slower exchange rate for DCS is consistent with a longer residency time on the enzyme. I would expect this to show up as tight binding behaviour. But instead, DCS is well known to be a traditional competitive inhibitor with low microM K_i. Can the authors comment on this apparent discrepancy?

The self resistance mechanism by the producing organism is postulated to be the result of production of DdIs with poor DCS affinity (JBC (2004) 279:46143-52). Can the authors use this information and sequence identity (or lack thereof) with DdlB to help support the proposed phosphorylation mechanism?

Ref 9 is not appropriate for line 106. As far as I know there is no reported PIX data for DdlB. The original work by Raushel and Walsh was reported for Salmonella DdlA (JBC (1990) 265:8993-8) and the Van ligase work was reported in Healy et al (2000) Chem Biol 7:505-14.

Reviewers' comments:

Reviewer #1 (Remarks to the Author):

The manuscript by Batson et al describes, by structural analysis and demonstration of trans-phosphorylation using the technique of positional isotope exchange, that the antibiotic, D-cycloserine, is phosphorylated by MgATP in the active site of the D-ala D-ala ligase (DdlB) of *E. coli*. The authors maintain that this is the first demonstration of this mode of inhibition by an antibiotic. The paper also suggests that the product, D-cycloserine phosphate (DCSP) should not bind to the NMDA receptor, as does D-cycloserine (DCS), the latter of which is neurotoxic. The unequivocal demonstration of DdlB-catalyzed formation of DCSP from MgATP, along with the attending structure DdlB-MgADP-DCPS, is an important finding, as it will influence the design of other "phosphorylatable" analogues of DCS.

However, this reviewer does not believe that this paper is suitable for publication in *Nature Communications* because there is no other detail about whether or not the phosphorylation of DCS has any kinetic significance in the inhibition of DdlB or that its phosphorylation creates a chemical species that is more inhibitory than DCS.

(1) For example, the apparently reversible formation of DdlB-MgADP-DCSP is far slower than D-Ala-D-Ala formation, that is to be expected, but does DCSP formation lead to time-dependent inhibition of DdlB on the same timescale as the rate of PIX, meaning that DCSP leads to a tighter-binding inhibitor than DCS? Or is PIX just an artifact of a long incubation of MgATP, DCS and very large (μM) levels of enzyme?

These are very good points. Firstly, time-dependent inhibition of *E. coli* DdlB and of *M. tuberculosis* Ddl have been recently reported by our group (Prosser GA, de Carvalho LP. *Biochemistry*. 2013 52(40):7145-9). When we wrote that manuscript we were unaware of the formation of DCSP, and therefore interpreted slow-binding kinetics as a simple slow binding event. In light of the results present here, we can now expand the mechanistic interpretation of slow-onset inhibition of D-Ala:D-Ala ligases, incorporating the phosphorylation of DCS. Secondly, PIX is not an artefact of potential "long incubation times with high enzyme concentration". If it was, it should increase with time on free enzyme and be suppressed by DCS. We actually see the opposite behaviour. PIX is negligible in the absence of DCS at the same enzyme concentrations and significant in the presence of DCS. Also, non-enzymatic hydrolysis of ATP under these conditions is irreversible, and therefore, cannot account for PIX, which is by definition reporting on "intact" ATP as oppose to ADP. We have amended the manuscript text to clarify these issues. It now reads: "No PIX reaction was observed in control experiments performed in absence of DCS at the same concentration of DdlB, confirming that DCS is the specific acceptor of the ATP γ -P group during the phosphate transfer reaction".

(2) The inhibition data shown in Figure 1g indicates an IC50 value of DCS-mediated inhibition of the ATPase activity of DdlB of 11.5 μM . How does this value compare with the competitive inhibition constant of DCS vs. D-ala under full biosynthetic reaction conditions?

The IC50 obtained for DCS-inhibition of ATPase activity (11.5 μM) is well in the range of Kis obtained for DCS-inhibition of *E. coli* Ddls biosynthetic reaction (9 μM for DdlA and 27 μM for DdlB) (Noda M et al., 2004, *JBC* 279: 46143-46152).

Is the rate of PIX that reaches a value of zero at 8 hrs due to the achievement of PIX equilibrium or has the formation of a tight-binding DdlB-MgADP-DCSP complex shut down the enzyme?

The rate of PIX at 8 hours is close to zero as the reaction is almost at equilibrium. The slow rate for attaining this equilibrium is caused by slow transfer of the phosphate from ATP to DCS, from DCSP back to ADP or both. As the reviewer pointed out before, it is not surprising that this process is slow, as DCS is

not the physiologic substrate. We have now amended the manuscript text to clarify that. It reads: "The PIX rate reaches a value of zero at around 8 hours, this is likely due to equilibrium of the reaction and the nature of the acceptor, DCS, which is not the physiologic substrate but instead an inhibitor".

If the value is the same at short incubation times with the full complement of substrates, than this suggests that the formation of DCSP provides no inhibitory benefit. It would be helpful to know if the inhibition of DdlB by MgATP + DCS and by MgADP + DCSP are comparable, or does the enzyme-catalyzed formation of DCSP provide an especially potent inhibition as seen for glutamine synthetase + MgATP + methionine sulfoximine.

We agree with the reviewer, that having those two inhibition constants would be great. However, as we (and others) have pointed out in the manuscript, DCSP is likely to be highly unstable in solution. Other groups, such as the Walsh group spend significant time and resources trying to make compounds such as DCSP, D-Ala-P and so on. They were convinced that such acyl-phosphates are highly unstable in aqueous solution. Therefore, we cannot obtain the two dissociation constants via separate methods.

(3) Is the formation of DdlB-MgADP-DCSP reversible, and thereby, is DdlB expected to create a pool of intra-bacterial DCSP that would perhaps trap DCSP within the bacterium and lessen the availability of DCS to exposure to neural NMDA receptor? This is a simple dialysis experiment to determine if DdlB activity may be restored after the formation of the DdlB-MgADP-DCSP ternary complex.

Formation of DCSP is unquestionably reversible, otherwise no PIX could be detected and is by definition an absolute requirement for a PIX effect. Now, as we pointed out in the revised version of the manuscript, DCSP is highly unstable in solution, in an analogous chemical manner to D-Ala-P, and therefore we do not expect the existence of a free DCSP pool.

(4) To point 3, if DCSP is metabolic insignificant, I do not see the value of the modelling studies demonstrating that DCSP should not bind to NMDA. And also, if the course of next-generation inhibitor design pursues the notion that analogues of DCSP are the path to proceed, it should first be shown that DCSP itself is a DdlB inhibitor. This has not been shown. DCSP could be made in micromolar amounts enzymatically.

Apologies that we did not make this point as clear as we needed. The modelling experiment was not meant to point to a potential way to use DCSP as a drug, as it is not stable in aqueous solution. The point of the modelling experiment is to show that if we, or other group, were able make a stable analogue of DCSP, which could still inhibit DdlB, this compound would be devoid of NMDA binding and hence NMDA-mediated CNS toxicity, which is by far the main problem faced when treating a patient with DCS. It is an obvious attractive way forward in terms of drug discovery, which can only be envisage in light of the results presented in this manuscript, i.e. formation of DCSP in Ddl enzymes. We have now amended the text to make this point abundantly clear.

(5) There is no description of crystallographic conditions for the DdlB-MgADP-DCSP that are salient, at least to this reviewer.

We apologies that this was not clear to the reviewer, we have altered the text of the manuscript so that it is now totally clear we refer to reference 11 as indicated in the original document.

(6) This reviewer believes that eq 1 is meant to be the Michaelis-Menten equation

We thank the reviewer for pointing this out and apologies for this oversight. The reviewer is correct and we have corrected the equation in the manuscript.

Reviewer #2 (Remarks to the Author):

In the manuscript entitled "Antibiotic inhibition of D-Ala:D-Ala ligase through a novel phosphoryl intermediate" by Batson et al. (Manuscript NCOMMS-16-28288-T) the authors describe a novel chemical entity, D-cycloserine-phosphate (DCSP) as inhibitor of E. coli D-alanine ligase. The observation of DCSP is a distinct phosphorylated form of D-cycloserine (DCS) generated by the transfer of the γ -5'-phosphoryl moiety of ATP to the 3'-oxygen of DCS. Later molecule is a commercial drug against multi-resistant M. tuberculosis strains.

The authors determined three crystal structures: DdlB co-crystallised with ADP and DdlB co-crystallised with D-Ala, ATP as well as DdlB co-crystallised with D-Ala-D-Ala, and ATP and DCS. The structural investigations are further supported kinetic measurements in particular ^{31}P NMR experiments to prove the existence of DCSP.

To the reviewer's point of view the current version of the manuscript is very well written and comprehensive.

We would like to thank reviewer 2 for their kind comments and suggestions.

- What is the sequence identity between E. coli and M. tuberculosis? Are key residues in the active site conserved or different?

That is a great question. The actual level of amino acid identity between E. coli DdlB and M. tuberculosis Ddl is about 27% identity as calculated over the 60 aa longer Mtb DdlB sequence from a structure-based alignment. However, there are a number of significant structural differences between the two enzymes, including a much longer N-terminus (1-112) and the active site (from 244 to 269) of the *M. tuberculosis* enzyme with long loop insertions (259-269 and 125-129), compared to E coli, which make them rather different. These include suggested E. coli key residue E15 which finds only a weak structurally-related counterpart in E123 of Mtb. In fact, to date there is only one crystal structure of M. tuberculosis Ddl, in an apo form, in sharp contrast with the large number of structures of Ddl enzymes from other sources with a number of ligands.

- Do the authors have an idea about the origin of the carbonate in vicinity of ADP and D-Ala-D-Ala?

This is an interesting question for which we have no direct answer! The density for the molecule between the ADP and dipeptide is planar and consistent with the existence of a carbonate molecule, possibly from dissolved CO_2 in the crystallisation buffer, that takes the position of the gamma phosphate of ATP.

- Are there differences in the active sites of both monomers in the dimer?

The active sites are essentially identical. Interestingly an indication that the enzyme may be characterized by some degree of cooperativity for the binding of DCS is indicated by a Hill number value of 1.50 obtained for the DCS inhibition of ATP hydrolysis. We have realised that we did not specify this value in the previous version of the text but have now amended commented on that. It now reads as: *Interestingly a value of 1.50 for the Hill number seems to suggest the existence of some degree of negative cooperativity in the binding of the inhibitor for the ATP hydrolysis reaction.*

Minor points:

- Figure 1: Why the authors choose a different color for the adenine and ribose function in panel b and c?

Panels A and B are designed with identical colours to enable the observer to directly compare the ATP and ADP bound states respectively. Panel C is separate and provides the electron density for the DCSP

bound form. We have now unified the colour scheme for all three panels and added the Mg²⁺ ions as below.

Maybe the Mg²⁺ ions could also be displayed in panel a and b.

Mg²⁺ ions have now been added

- SI Table 1: It would be beneficial to report CC_{1/2} for the experimental data
- Frequently spaces are missing between value and unit.

We have added these values to the table SI table 1 as instructed

Reviewer #3 (Remarks to the Author):

This report describes a re-evaluation of the mechanism of an old natural product antibiotic, D-cycloserine (DCS), on one of its molecular targets, D-Alanine-D-Alanyl Ligase (Ddl), an enzyme essential for bacterial peptidoglycan biosynthesis. The first reports of DCS inhibition of Ddl dates to 1964 in pioneering work by Neuhaus. DCS also inhibits the preceding enzyme in the biosynthetic pathway, Alanine racemase. Blocking more than one biosynthetic step makes DCS an attractive clinical candidate. Unfortunately, DCS also targets the NMDA receptor, resulting in challenging toxicity issues and diminishing DCS's potential as a useful drug (though it does find use as a back up drug for Mtb infection).

The authors demonstrate, using 3D-structure determination and kinetic PIX experiments, that DCS can be phosphorylated by DdlB, generating a new phosphorylated species. Phospho-DCS is expected (but not explicitly shown) to be unstable, and finding it in the enzyme active site, associated with residues known to stabilize alanine-phosphate, suggests an active site environment that can accommodate phosphor-DCS-like compounds. Using modeling, the authors predict that this phosphorylated species should not bind to the NMDA receptor and therefore new chemical species that mimic this structure should retain enzyme inhibition (and therefore perhaps antibiotic?) activity while diminishing NMDA-associated toxicity.

The finding that DCS is phosphorylated is I believe completely unexpected and unprecedented. This antibiotic has been studied in academic and pharma labs for over 50 years with no hint that this phospho-DCS was a potential intermediate. The crystal structure is unambiguous and the PIX data are consistent with the proposed mechanism. Overall this is an important addition to the antibiotic literature.

We would like to thank the reviewer for her/his encouraging words and comments on our manuscript pointing out the finding that DCS is phosphorylated is unexpected and unprecedented.

How hard did the authors try to isolate phospho-DCS? I expect while the half life may be fast, it should be detectable.

We have tried long and hard to detect this species! We tried to detect it by mass spectrometry and NMR spectroscopy under a variety of conditions as well as a variety of chemical trapping methods. Based on precedent with D-Alanyl-P, we believe the half-life to be too short to permit any isolation of such acyl phosphates in aqueous media. This correlates with similar but ultimately unsuccessful efforts made by the group of Professor Chris Walsh in their paper Healy et al 2000 Chem Bio & p505-14.

I am puzzled by some of the kinetic data. The PIX rate was determined to be 18.5 min⁻¹. I'm not aware of the D-Ala PIX rate for DdlB, but the rate for Van enzymes is <1 min⁻¹ (Healy et al (2000) Chem Biol 7:505-14). The slower exchange rate for DCS is consistent with a longer residency time on the enzyme. I would expect this to show up as tight binding behaviour. But instead, DCS is well known to be a traditional competitive inhibitor with low microM Ki. Can the authors comment on this apparent discrepancy?

It was surprising for us as well as first, but this is PIX for an enzyme-inhibitor complex. This is the first time that such experiment has been performed and reported, that we are aware of. We believe the slow PIX rate in this case to be caused by the fact that DCS is not the physiologic substrate. In addition, as we mentioned earlier on in this revision, we have demonstrated that DCS is in fact a slow-onset inhibitor of Ddl enzymes (Prosser GA, de Carvalho LP. *Biochemistry*. 2013 52(40):7145-9). Time-dependent inhibition is barely seen using manual mixing but very apparent when the experiments were carried out with automatic mixing, in a stopped-flow. Also, time-dependent inhibition is more apparent with the *M. tuberculosis* enzyme.

The self resistance mechanism by the producing organism is postulated to be the result of production of

Ddls with poor DCS affinity (JBC (2004) 279:46143-52). Can the authors use this information and sequence identity (or lack thereof) with DdlB to help support the proposed phosphorylation mechanism?

We thank the referee for this suggestion. Our understanding of this manuscript was that K_i for DCS against Ddl was significantly different, by 40-100 fold. This in no way indicates that there is or there is no formation of DCSP in the enzyme from *S. lavendulae*. Also, the authors report that alanine racemase, the other target of DCS in bacteria is also differentially inhibited by DCS. Finally, it is important to note that one of the main mechanisms for "self-resistance" in bacteria that produce antibacterial agents is by pumping them out of the cell very efficiently. The authors did not address this key point in their JBC paper referenced. This is of course important, because if the K_i is up by 100-fold but the intrabacterial concentration of DCS is up by 1000-fold, these enzymes should be fully inhibited.

Ref 9 is not appropriate for line 106. As far as I know there is no reported PIX data for DdlB. The original work by Raushel and Walsh was reported for Salmonella DdlA (JBC (1990) 265:8993-8) and the Van ligase work was reported in Healy et al (2000) Chem Biol 7:505-14.

Apologies for this oversight which we have now corrected with both references to PIX as described above

Reviewers' comments:

Reviewer #1 (Remarks to the Author):

Having looked over the revisions to the manuscript by Batson et al, the points raised in my review have not been adequately addressed. If the authors were to revise their manuscript further, I feel it would be acceptable for Nature Communications, but not so in its current form.

The questions raised were these:

(1) is the observed rate of positional isotope exchange (PIX) equal or comparable to the observed rate of the establishment of a tightly-bound E-MgADP-DSCP complex? If so, then the transfer of phosphate from ATP to DCS and a subsequent conformation change may be the mechanism by which tight-binding is achieved. It seems to me this is an easy question to address since data should exist for both kinetic rates, PIX in this paper, and the rate of onset of tight-binding inhibition as eluded to by the citation of Prosser and de Carvalho.

(2) Is the rate of PIX that reaches a value of 0 at 8 hours due to inactivation of the enzyme due to tight-binding or has the $^{18}O_4$ simply been exchanged as far as it can go? Has equilibrium of exchange truly been achieved or is the E-ADP-DCSP complex at a conformational dead end? The addition of fresh $[^{18}O_4]ATP$ to the E-ADP-DCSP mixture, with re-institution of new PIX would address this question. or...

(3) Another aspect of question (2) is whether or not dialysis of the E-ADP-DCSP complex would restore free enzyme or is this ternary complex kinetically irreversible. In other words, does PIX occur up to a point where the E-ADP-DCSP complex may either go backward to release DSC and ATP or progress to a conformationally irreversible complex: $E^*-ADP-DCSP$ that leads to cessation of PIX? What evidence was provided that DSCP is highly unstable in solution as cited by the authors in the revised manuscript?

(6) In the revised manuscript, equation (2) reads:

$v = V_{max} \times A / (K_m - A)$. There is an error in the demoninator

Reviewer #3 (Remarks to the Author):

All the issues I raised have been addressed. I think this is a fascinating result.

Reviewer #1 (Remarks to the Author): Having looked over the revisions to the manuscript by Batson et al, the points raised in my review have not been adequately addressed. If the authors were to revise their manuscript further, I feel it would be acceptable for Nature Communications, but not so in its current form.

The questions raised were these:

(1) is the observed rate of positional isotope exchange (PIX) equal or comparable to the observed rate of the establishment of a tightly-bound E-MgADP-DSCP complex? If so, then the transfer of phosphate from ATP to DCS and a subsequent conformation change may be the mechanism by which tight-binding is achieved. It seems to me this is an easy question to address since data should exist for both kinetic rates, PIX in this paper, and the rate of onset of tight-binding inhibition as eluded to by the citation of Prosser and de Carvalho.

The two rates in question do not need to be identical as one was obtained in the presence of D-Ala, which is competitive against DCS and the other in the absence. We would like to refrain from making this comparison as we do not believe we have enough information to conclude that we are in fact looking at the same exact process in these two studies. Also, most of the experiments carried out by Prosser and Carvalho were carried out under conditions where site 2 inhibition was forced, which clearly does not apply to the PIX experiment, where DCS is by definition binding to site 1.

(2) Is the rate of PIX that reaches a value of 0 at 8 hours due to inactivation of the enzyme due to tight-binding or has the $^{18}\text{O}_4$ simply been exchanged as far as it can go? Has equilibrium of exchange truly been achieved or is the E-ADP-DCSP complex at a conformational dead end? The addition of fresh $^{18}\text{O}_4$ ATP to the E-ADP-DCSP mixture, with re-institution of new PIX would address this question. or... ☐

Following to the referee's suggestion we have performed the control experiment where, after running the PIX reaction for 20h, we have added fresh $^{18}\text{O}_4$ ATP and observed identical exchange kinetics for the following 20 h, indicative that a true isotopic equilibrium is reached in the PIX reaction rather than enzyme inactivation or denaturation. The control experiment is now shown in the Supplementary Figure S3, right panel.☐

(3) Another aspect of question (2) is whether or not dialysis of the E-ADP-DCSP complex would restore free enzyme or is this ternary complex kinetically irreversible. In other words, does PIX occur up to a point where the E-ADP-DSCP complex may either go backward to release DSC and ATP or progress to a conformationally irreversible complex: E*-ADP-DCSP that leads to cessation of PIX? What evidence was provided that DSCP is highly unstable in solution as cited by the authors in the revised manuscript? ☐

We thank the referee for her/his comments in respect of this clarification and her/his question regarding the dialysis of the enzyme in relation to restoration of "*free enzyme or is this ternary complex kinetically irreversible?*".

To be rigorous in our reply, we performed an experiment in which DdlB was incubated with an excess of ATP and DCS prior to rapid remove of the remaining free ligand using a fast desalting column into fresh enzyme buffer followed by immediate kinetic analysis. When the enzyme was removed from an excess of DCS and ATP it regained 100% of the WT activity as expected as discussed above. This is inconsistent with an kinetic irreversible E:DCSP:ADP complex.

In addition, prompted by the referee's question on DCSP instability, we have also performed an experiment with unlabeled ATP in conditions identical to the PIX experiment, except that the enzyme concentration was 100 μ M. Potential formation of DCSP was evaluated by high resolution mass spectrometry. We were unable to detect any DCSP species, which if present in one-to-one stoichiometric ratio with the enzyme would have been at a final concentration of 25 μ M in the LCMS sample.

In respect of the referees comments on "evidence was provided that DSCP is highly unstable in solution as cited by the authors " We thank the referee for this comment and have added additional references as summarised below. All previous studies on acyl phosphate intermediates have underlined the instability of these species in aqueous solution as well as the difficulty of actually isolating and characterising these species. In addition to the studies from Walsh et al indicated in the paper cited in reference 12 of the manuscript we have added the following text to clarify the situation regarding acyl phosphate instability:

The highly reactive nature of acyl phosphates, and in particular, the great difficulties in terms of isolating and characterising them, has been well established. Although very early chemical approaches have shown that insoluble silver salts of acetyl phosphate itself can be isolated [e.g. see Bentley, R. J. Am. Chem. Soc., 1949, 71, 2765 – 2767], all other reports involving these systems indicate that they are reactive and subject to ready decomposition. For example, under neutral (pH = 7.2) conditions, it has been shown that acyl phosphates are rapidly hydrolysed (e.g. rate of hydrolysis of acetyl phosphate at 39 °C and pH 7.2 = $4.4 \times 10^3 \text{ min}^{-1}$; see G. Di Sabato and W.P. Jencks, J. Am. Chem. Soc, 1961, 83, 4400 -4405). More recent studies (Biron, J-P., and Pascal, R. J. Am. Chem. Soc., 2004, 126, 9198 – 9199) report that acyl-phosphorylated amino acids (e.g. that derived from valine), generated in aqueous solution, can be detected using NMR, but are somewhat transient, and undergo steady decomposition (rate = $5.7 \times 10^{-4} \text{ s}^{-1}$, corresponding to a halving of the concentration of the acyl phosphate intermediate after 30 mins). These authors briefly investigated the mechanism of decomposition of the acyl phosphates and report that addition of methanol to the aqueous solution followed by evaporation of the solvents allowed the identification of methanol cleavage products (methyl phosphate) corresponding to attack of the phosphate group in the intermediate amino acid acyl phosphate by methanol.

In the revised manuscript, equation (2) reads: $v = V_{\text{max}} \times A / (K_m - A)$. There is an error in the demoninator

Thank you for highlighting this typo which has now been corrected.